# Unlicensed Molnupiravir is an Effective Rescue Treatment Following Failure of Unlicensed GS-441524-like Therapy for Cats with Suspected Feline Infectious Peritonitis

**DOI:** 10.3390/pathogens11101209

**Published:** 2022-10-20

**Authors:** Meagan Roy, Nicole Jacque, Wendy Novicoff, Emma Li, Rosa Negash, Samantha J. M. Evans

**Affiliations:** 1Department of Veterinary Biosciences, College of Veterinary Medicine, The Ohio State University, Columbus, OH 43210, USA; 2Independent Researcher, San Jose, CA 95123, USA; 3Departments of Orthopaedic Surgery and Public Health Sciences, School of Medicine, University of Virginia, Charlottesville, VA 22903, USA

**Keywords:** FIP, coronavirus, antiviral, EIDD-2801, black market

## Abstract

Feline infectious peritonitis (FIP) is a complex and historically fatal disease, though recent advances in antiviral therapy have uncovered potential treatments. A newer therapeutic option, unlicensed molnupiravir, is being used as a first-line therapy for suspect FIP and as a rescue therapy to treat cats who have persistent or relapsed clinical signs of FIP after GS-441524 and/or GC376 therapy. Using owner-reported data, treatment protocols for 30 cats were documented. The 26 cats treated with unlicensed molnupiravir as a rescue therapy were treated with an average starting dosage of 12.8 mg/kg and an average ending dosage of 14.7 mg/kg twice daily for a median of 12 weeks (IQR = 10–15). In total, 24 of 26 cats were still living disease-free at the time of writing. One cat was euthanized after completing treatment due to a prolonged seizure, and the other cat underwent retreatment for relapsed clinical signs. Few adverse effects were reported, with the most notable—folded ears (1), broken whiskers (1), and severe leukopenia (1)—seen at dosages above 23 mg/kg twice daily. This study provides a proof of principle for the use of molnupiravir in cats and supports the need for future studies to further evaluate molnupiravir as a potentially safe and effective therapy for FIP.

## 1. Introduction

Feline infectious peritonitis (FIP) is a complex and historically fatal disease caused by a mutation of the ubiquitous feline enteric coronavirus (FECV) [1]. Recent advances in feline and antiviral medicine have uncovered potential treatments for FIP. The 3C-like-protease inhibitor GC376 was the first targeted antiviral therapy used against this disease [2]. GC376 was highly effective at improving the clinical signs of FIP in 19 out of 20 naturally infected cats but showed a limited ability to control long-term disease [2]. Pedersen et al. continued to investigate the antiviral compound GS-441524, a nucleoside analogue and active metabolite of remdesivir (GS-5734). GS-441524 demonstrated a superior ability to treat and control disease in naturally infected cats compared to GC-376, with 25 out of 31 cats being disease-free at the time the report was written [3].

Since these discoveries, cat owners around the globe have been obtaining these mostly unlicensed drugs to treat their FIP cats with remarkably high success [4]. In the United States, there is high demand for legal FIP therapy due to the ethical and legal concerns surrounding unlicensed GC376 and GS-441524. Additionally, some cats with FIP have exhausted all current treatment options due to disease relapse and/or treatment failures after GS-441524, GC376, and/or combination therapy. Thus, there is an urgent need for an effective, legal treatment option for FIP.

With the recent SARS-CoV-2 outbreak, numerous novel antivirals have been brought onto the market. Molnupiravir (EIDD-2801), manufactured by Merck, is currently available under emergency use authorization (EUA) by the FDA to treat COVID-19 in adults [5]. It is an oral prodrug of the nucleoside analog B-D-N4-hydroxycytidine that increases guanine to adenine and cytosine to uracil nucleotide transition mutations in coronaviruses [6]. This mechanism increases the mutation rate beyond the accepted threshold, subsequently inactivating the virus [7]. Molnupiravir has been found to be safe and well-tolerated at a dose of up to 800 mg twice daily in COVID-19 patients [8]. Some studies have reported a drastic decrease in hospitalization rates and death for COVID-19 patients with mild to moderate disease, though the efficacy appears to be lacking for COVID-19 patients with severe disease [7]. 

Due to the strong potential of molnupiravir to treat other coronavirus infections, cat owners have been using unlicensed molnupiravir (or its active metabolite, EIDD-1931) purchased over the internet as a therapy for FIP. However, the use of molnupiravir for the treatment of FIP is not currently documented in the scientific literature. Unlicensed molnupiravir is being used as a first-line therapy for suspected FIP and as a rescue therapy to treat cats who have persistent or relapsed clinical signs of FIP after GS-441524 and/or GC376 therapy. The purpose of this study is to document that usage and provide a proof of principle for molnupiravir as a potential therapeutic for FIP, according to owner-reported data.

## 2. Materials and Methods

A survey was created using Qualtrics XM (Qualtrics Version May–August 2022, Provo, UT, USA) under the Ohio State University’s license. The survey (Appendix A) was written in English and consisted of 94 multiple choice and free response questions asking about FIP diagnosis, clinical signs, initial therapies (used prior to molnupiravir), molnupiravir therapy, adverse events, the duration of treatment, and remission times. The number of free response questions was limited to reduce recall bias. The survey also allowed owners to upload relevant documents (e.g., veterinary medical records and laboratory results). The survey was formatted using questions from previous studies, to remain consistent in language and style, as well as newly created questions specific to the molnupiravir therapy experience. Survey logic dictated that some questions only appeared after a particular answer was selected, while others were skipped if a particular answer was selected. This conditional logic was used to reduce reporting bias and survey-taking fatigue. The survey took approximately 20–30 min to complete and could be saved and completed later, if needed. This study was approved by the Ohio State University Institutional Review Board (protocol #2021E0162).

The survey was distributed by email individually to participants, and data were collected from June to August 2022. Participants were identified from a subset of owners seeking molnupiravir therapy for their cat with suspected FIP through popular FIP therapy and support groups on social media. Inclusion criteria were surveys regarding cats presumed to have FIP based on diagnostics from a veterinarian, failure to respond to initial therapy or relapse of clinical signs after the completion of an initial therapy other than molnupiravir (e.g., GS-441524 or GC376), and the completion of 8–10 weeks of orally administered molnupiravir therapy (or those who subsequently died or were euthanized during therapy). A small cohort of cats who received molnupiravir for 8–10 weeks as the initial and only therapy, to be referred to as first-line therapy for the remainder of the paper, for suspect FIP were also included in this study. Exclusion criteria were surveys with incomplete data or cats that were not diagnosed with FIP by a veterinarian.

## 3. Results

### 3.1. Demographics

A total of 80 potential participants were identified through the social media FIP support group, and 37 survey invitations were sent to those participants with available contact information. In total, 33 surveys were submitted, and follow-up emails were sent to 21 participants in order to obtain complete survey data. Seventeen owners attached relevant documents to their survey submissions, and two additional owners emailed relevant documents to the study’s email account, which included veterinary medical records, laboratory results, and diagnostic imaging. These aforementioned documents were used to support the adverse reactions reported by one participant. One response was to decline to participate. Two cases were excluded because the cats did not receive a diagnosis of FIP from a veterinarian (one was reportedly self-diagnosed following the loss of a sibling to FIP, and the other was seen by a veterinarian who concluded that bloodwork was not consistent with FIP). Thus, a total of 30 cats with suspect FIP were included in this study, 4 of which did not receive any therapy prior to molnupiravir. These four cats were included as a separate small cohort of first-line molnupiravir treatment. A flowchart of these cases is shown in Figure 1. The represented countries of origin were the United States (25), Germany (2), Poland (2), and Sweden (1). The sex/neuter status of the cats at the time of diagnosis was 40% neutered male, 40% spayed female, and 20% non-neutered male. The mean age at diagnosis was 9.7 months of age, with a range from 1 month to 6 years. Most cats were of mixed or unknown breed (70%); there were seven purebred cats and two designer mixed breed cats (e.g., a Balinese/Ragdoll mix and a Siamese mix). Responses indicating a cat as an “American Shorthair” or “American Longhair” were instead categorized as mixed breed, due to the commonly observed confusion of American owners pertaining to this breed’s nomenclature.

With regard to comorbidities, only one cat was reported to have feline leukemia virus, and one other cat was reported to have calicivirus. Low numbers also reported a history of external and/or internal parasitic infections (3), conjunctivitis/ocular infections (2), and bacterial skin infections (pyoderma) (1). A total of 16 cats were reported to have neurological manifestations of FIP. Three were reported to have both neurological and ocular manifestations of FIP, and two reported only ocular FIP. Of the remaining cases, seven were effusive, while five cases were noneffusive. A complete breakdown of the FIP types is shown in Table 1.

### 3.2. Initial Therapy Prior to Molnupiravir

In total, 26 out of 30 cats received an initial treatment for suspect FIP with unlicensed GS-441524 or a drug combination including unlicensed GS-441524 as the main base drug (GS-441524-based). Half (13) of the cats were treated with injectable GS-441524. Only three cats were treated with oral GS-441524, with an additional seven treated with a combination of injectable and oral GS-441524 throughout the duration of therapy. Two were treated with a combination of unlicensed GS-441524 and unlicensed GC376. Cat #6 was treated with all the previously mentioned drugs along with molnupiravir for 12 weeks of a highly complicated regimen (Appendix A). The dosages for the combination drugs used as part of the first-line therapy (e.g., GC376 and molnupiravir) were not collected. The reported starting dosages for unlicensed GS-441524 ranged from 2 mg/kg to 10 mg/kg; the most commonly reported were 5–6 mg/kg (eight cats) and 10 mg/kg (seven cats). Most (21) cats were dosed once a day. Only four were dosed twice a day, and one cat was initially dosed twice a day for one week before switching to once a day. The median duration of GS-441524-based therapy was 12 weeks (IQR = 12–13). Fifteen cats were reported to have changed daily doses during the treatment duration. Several cats increased the daily dose according to body weight to maintain the same dosage in mg/kg. Others increased the dosage in mg/kg due to a lack of clinical response or a change in the route of administration (e.g., injectable to oral GS-441524). No decreases in dose over the duration of therapy were reported by any participants.

A total of 6 out of 26 cats received less than the median 12-week duration of GS-441524-based therapy due to a lack of clinical response and immediately began a different course of treatment. Two out of six cats began another route or dosage of treatment with unlicensed GS-441524, as shown in Table 1. One cat switched from injectable to oral GS-441524 for the second treatment. The other cat simply increased the dosage of GS-441524 for the second treatment. The remaining four cats began treatment with unlicensed molnupiravir at this time, as seen in Table 2. Of the 20 cats who completed at least the 12-week duration of GS-441524 therapy, 16 were reported to have reached a clinical remission. All 16 were in remission for less than 6 months, with 2 cats in remission for less than a week prior to the return of clinical signs. All 16 began a second round of therapy, with 10 receiving a second round of GS-441524-based therapy and 6 starting molnupiravir at this time. The four cats who completed the course of GS-441524 but did not reach clinical remission immediately began molnupiravir therapy. In summary, 26 cats received GS-441524-based first-line therapy, and all 26 relapsed or did not respond adequately. In total, 10 out of 26 received a second round of GS-441524-based therapy, and 16 began molnupiravir.

### 3.3. Second Therapy Prior to Molnupiravir

In total, 10 of the 26 cats who received initial GS-441524 therapy and subsequently relapsed were reported to have received a second round of unlicensed GS-441524 before starting molnupiravir therapy. Again, most cats received injectable GS-441524 (6), with two receiving oral GS-441524 and two receiving both injectable and oral GS-441524. The reported dosages ranged from 4–5 mg/kg to 15 mg/kg; the most common dosages used were 7–8 mg/kg (two cats) and 15 mg/kg (two cats). Most cats were treated once a day (seven cats), with one cat dosed twice a day and one cat dosed three times a day. Most cats were reported to have had dose changes over the duration of therapy. Two doses were adjusted with weight gain to maintain the same mg/kg dosage. The dosage in mg/kg was increased for five cats who were not responding adequately or had new clinical signs (e.g., neurological signs). 

The median duration of therapy was 12.5 weeks (IQR 9.75–14.25). Only two cats did not receive at least 12 weeks of therapy. One of the two added GC376 and molnupiravir to the current GS-441524 therapy, and the other started molnupiravir as a sole therapy. Of the eight cats who completed at least a 12-week duration of GS-441524 therapy, two did not report reaching clinical remission. Both cats began unlicensed molnupiravir therapy at that time. The remaining six cats were reported to have reached clinical remission after the second round of GS-441524 therapy. Five of the six cats were in remission for less than 4 weeks, the exception being one cat who was in remission for more than 6 months but less than a year. Seven out of ten cats began unlicensed molnupiravir at this time. 

### 3.4. Third Therapy Prior to Molnupiravir

The three remaining cats received a final round of GS-441524-based therapy before switching to molnupiravir therapy. Cat #2 received oral GS-441524 for 5 weeks before beginning molnupiravir. Cat #9 received 6 weeks of oral and injectable GS-441524 and then continued with 6 weeks of only oral GS-442524. The dosage and frequency for both cats are unknown, as the survey only collected data for two therapies prior to molnupiravir. Cat #21 received a combination of GS-441524, GC376, and molnupiravir for 12 weeks. The dosage, frequency, and duration of each changed drastically over 12 weeks (Appendix A). All three cats began molnupiravir with no clinical remission from this third round of therapy.

### 3.5. Molnupiravir as Rescue Therapy

Of the 26 cats receiving unlicensed molnupiravir as rescue therapy, most used an Aura brand of the medication, with only 2 reported to use a different brand of molnupiravir. Over 81% of the cats (18) were treated with Aura 2801, 1 cat was treated with Aura 1931, and 2 other cats were treated with both Aura formulations. The average starting dosage was 12.8 mg/kg twice a day. One cat was dosed only once a day, and two cats were dosed 2 to 3 times a day. The most common starting dosage used was 12 mg/kg twice a day. The dosage ranged from 6 to 28 mg/kg twice a day. There were 11 dosage changes reported, with all but one being a dosage increase. No explanation was given for the dosage decrease for cat #3. The average ending dosage was 14.7 mg/kg twice a day, with the same three cats differing in the frequency of dosing. The most common ending dosage was also 12 mg/kg twice a day. The dosage range was 7 to 30 mg/kg twice a day. 

The median duration of therapy was 12 weeks (IQR 10–15). There was a wide reported range overall of 7–20 weeks. Only eight cats received less than 12 weeks of treatment. The cat who only completed 7 weeks of therapy was reported to have ceased therapy due to reaching clinical remission. All 26 cats completed a duration of therapy of 7 weeks or greater, and all 26 cats survived. There were no reports of missed doses of molnupiravir therapy. 

Owners reported the improvement of clinical signs in over 92% of cats within three weeks of commencing molnupiravir therapy, with 84.6% of cats reported to display improvement within two weeks and almost half (46.2%) within one week. Only two cases were reported differently, with one cat not exhibiting any signs of improvement until 1.5 months and the owner of the other cat being unsure of the timeline and the degree of improvement of clinical signs. A total of seven cats were reported to have persistent clinical signs of FIP. One was reported to have a resolution of clinical signs after one week of the observation period. The remainder are thought to be residual, such as difficulty walking or jumping, tremors, MRI changes, and fecal incontinence. The full extent of the persistent clinical signs is found in Table 2. Only three cats were reported to have experienced adverse effects in response to molnupiravir, including nausea/vomiting, anorexia, folded ear tips (Figure 2), brittle whiskers, leukopenia, flaky skin, and muscle wasting. At the time of publication, 24 of 26 cats are living in clinical remission of FIP after oral molnupiravir therapy. One cat reportedly passed away one week after completing molnupiravir therapy due to a prolonged seizure, and the other cat (#21) was disease-free for 4 weeks before relapsing. Cat #21 was then started on a second round of molnupiravir at the same dosage but was subsequently euthanized due to a lack of response to treatment.

Cat #22 was reported to have severe leukopenia. Through veterinary records, cat #22 was found to have moderate panleukopenia with lymphopenia, neutropenia, and normal hemogram and thrombogram on 4 out of 5 sequential complete blood counts, which were confirmed through veterinary records of sequential complete blood counts. The first reported white blood cell count was 10,700 cells per microliter (reference range 3500–16,000 cells per microliter). The next four complete blood counts showed white blood cell counts ranging from 1200 to 1900 cells per microliter. The first neutrophil count was 8560 cells per microliter (reference range 2500–8500 cells per microliter). The next four neutrophil counts ranged from 696 to 1292 cells per microliter. The first lymphocyte count was 1177 cells per microliter (reference range 1200–8000 cells per microliter). The next four lymphocyte counts ranged from 330 to 532 cells per microliter.

### 3.6. Molnupiravir as a First-Line Therapy

A small cohort of four cats was treated with unlicensed molnupiravir as a sole therapy for suspect FIP, as shown in Table 3. Three reportedly chose molnupiravir over the unlicensed GS-441524 counterpart due to financial constraints. Cat #29 received a 12-week treatment of 12 mg/kg oral molnupiravir twice daily prior to the one shown in Table 3. This cat was disease-free for less than one week before restarting oral molnupiravir at 19 mg/kg twice daily for 10 weeks. 

All four cats were treated with Aura 2801 oral molnupiravir with an average starting dosage of 11.75 mg/kg twice daily (range 8–19 mg/kg) and an average ending dosage of 12.25 mg/kg twice daily (range of 8–19 mg/kg). The median duration of therapy was 11.5 weeks (IQR 10–13), with two cats being treated for 10 weeks, and two cats being treated for 13 weeks. A Mann–Whitney test was performed and found no significant difference between the median duration of therapy for molnupiravir as a rescue therapy (12) and the duration of therapy for molnupiravir as an initial therapy (11.5) (*p* = 0.692). All owners reported seeing clinical improvement within two weeks, and one cat appeared to improve within one week. All cats survived treatment, were disease-free at the time of publication, and did not report adverse effects of the treatment.

### 3.7. Molnupiravir Based on Type of FIP

The above information was combined for all 30 cats and then further separated according to the clinical forms of FIP. The 16 cats reported to have the neurological form of FIP were considered first. Then, the remaining cats were separated by ocular (2), effusive (7), and noneffusive (5) forms. The average starting dosage of molnupiravir for neurological FIP was 14.4 mg/kg twice daily, where two cats were treated 2–3 times a day. The average ending dosage was 16.4 mg/kg twice a day, again with two cats being treated 2–3 times a day. The most commonly used starting and ending dosage was 12 mg/kg twice a day. The median duration of treatment for neurological FIP was 12 weeks (IQR 10–12.641). 

For the two remaining ocular FIP cases, the average starting dosage was 11 mg/kg twice daily, and the average ending dosage was 13.5 mg/kg twice daily. They were treated for an average of 16.5 weeks. The seven effusive cases were treated with an average starting dosage of 10.5 mg/kg twice daily and an average ending dosage of 11.1 mg/kg twice daily. They were treated for a median of 13 weeks (IQR 12–16). The five noneffusive cases were treated with an average starting dosage of 10.6 mg/kg twice daily and an average ending dosage of 12.8 mg/kg twice daily. One cat was treated once daily. The median duration of therapy was 10 weeks (IQR 8.5–13.5).

### 3.8. Cost and Owner Satisfaction

Most cats in this study were switched to unlicensed molnupiravir therapy due to treatment failure/relapse or lack of response. In addition to cats relapsing or not responding to unlicensed GS-441524-based therapy, one cat was reported to not tolerate the injectable form of GS, and three owners were limited due to the cost of therapy. Owners were not required to disclose the financial cost of treatment; the following information was disclosed on a voluntary basis only. Additionally, responses of “0” that were reported were not included when calculating the following averages due to the inability to decipher whether a “0” meant no cost or an unknown cost. The average reported cost for the initial round of GS-441524-based therapy was USD 3448.83, and similarly the average reported cost for the second round of GS-441524-based therapy was USD 3509.09. Only 4 owners reported paying for molnupiravir therapy, while 16 others reported “0” (or no cost/cost unknown). The overall average for the 20 who recorded a response to the financial cost survey question (including responses of “0”) for molnupiravir was USD 209. The average cost for the four owners who did not respond with “0” was USD 1045. While 90% of owners reported being “extremely” or “somewhat” satisfied with their experience treating their cat with molnupiravir, three were “extremely dissatisfied” with their experience. Unfortunately, no explanation for the reported dissatisfaction was given.

## 4. Discussion

Here, we report the first known usage of unlicensed molnupiravir to treat suspected FIP in cats, according to owner-reported data. To treat cats using unlicensed molnupiravir as a first-line therapy for suspected FIP, a dosage of 12 mg/kg twice daily for approximately 12 weeks appears to be effective at achieving clinical remission, according to the combined data from this study. To treat cats using molnupiravir as a rescue therapy for failure or relapse after GS-441524-based therapy, a dosage of 12–15 mg/kg twice daily for 12–13 weeks appears to be effective at achieving clinical remission, according to the combined data from this study. However, when broken down by the clinical form of FIP, it was found that the neurological cases of FIP were generally treated at a higher dosage than the average for all types of FIP. The ocular, effusive, and noneffusive cases were treated with dosages around 12 mg/kg twice daily, with some variation. Therefore, a 15 mg/kg dosage of molnupiravir twice daily for 12 weeks appears to be effective for neurological cases of FIP. For ocular, effusive, and noneffusive cases, a 12 mg/kg dosage of molnupiravir twice daily for 12–13 weeks appears to be effective.

These data somewhat contrast with a proposed treatment protocol from a company manufacturing unlicensed molnupiravir under the brand name HERO Plus 2801. The package insert’s recommended dosages are 25 mg/kg once daily for effusive and noneffusive FIP, 37.5 mg/kg once daily for ocular FIP, and 50 mg/kg once daily for neurologic FIP [9]. The HERO Plus 2801 package insert also reveals preliminary data for “Oral Nutrition Treatment Effects on the Survival Time and Quality of Life for Feline Infectious Peritonitis”, a white paper that included 286 cats diagnosed with FIP. According to this package insert, 28 cats were cured with 4 weeks of treatment and 258 cats were cured after 8 weeks of treatment, with no deaths at the time of reporting [9]. The data from this study have yet to be published in the scientific literature.

Cats in this study, however, used molnupiravir from another supplier, Aura, who did not make specific treatment recommendations. The treatment protocols used were therefore based on the advice and information shared in social media groups, white papers published on the internet [10,11], and information about potential adverse effects contained in information published as part of human drug approval applications [12].

The molnupiravir treatment protocol deduced from this study aligns closer to an independently proposed protocol [10] published as a white paper online. By using data from in vitro cell cultures of EIDD-1931 and EIDD-2801, laboratory and field studies of GS-441524, and pharmacokinetic studies on people, these authors extrapolated an effective dosage for oral molnupiravir [10]. Their calculations suggested a dosage of 4.5 mg/kg every 12 h for effusive and noneffusive FIP, 8 mg/kg every 12 h for ocular FIP, and 10 mg/kg every 12 h for neurologic FIP [10]. Although the dosages in this study were generally higher than those proposed by the aforementioned authors, the high survival rate and low relapse rate at the time of survey completion suggest that unlicensed manufacturer recommendations may not represent the lowest effective dosage. Ultimately, controlled scientific experiments are badly needed to evaluate the lowest effective dosage of molnupiravir in cats with suspected FIP.

Several cats reported treatment with Aura 1931, which is the active metabolite of molnupiravir, EIDD-1931. The reported dosages used were similar in range to those reported for molnupiravir. Nominally, since the molecular weight of EIDD-1931 is lower than that of EIDD-2801, these cats received more of the active drug than cats using molnupiravir. However, a previous study showed decreasing oral bioavailability with increasing doses in mice. Therefore, the difference in bioavailable may not be proportional [13]. Unfortunately, feline pharmacokinetic studies of both molnupiravir and EIDD-1931 are lacking.

No adverse effects were reported on the HERO Plus 2801 package insert, contrasting what was reported in this study. Among the reported adverse effects of molnupiravir, the most notable were folded ears, losing whiskers, and severe leukopenia. No dermal or follicular lesions have been reported in the human medical literature to correspond with the whisker loss or ear folding reported here. However, it should be noted that the cats who experienced these side effects were receiving the two highest doses of molnupiravir reported in this study: 23 mg/kg three times a day and 30 mg/kg twice a day, respectively.

Severe bone marrow toxicity was reported in dogs during a 28-day trial that was cut short due to the severe effects of the drug [12]. All hematopoietic cell lines were reported to be affected at dosages of 17 mg/kg/day and 50 mg/kg/day [12]. Cat #22 received a maximum dosage of 23 mg/kg three times a day, which was much higher than the toxic dosage in dogs of 17 mg/kg once a day. The potential for reversibility was noted in the 17 mg/kg study group if treatment was halted [12].

There is concern about the contents of the unlicensed brands of molnupiravir, as these brands are not currently regulated and are frequently not labeled with their actual contents. The Hero brand (same manufacturer as HERO Plus 2801) shown in Figure 3 was analyzed by our group in December 2021 through Toxicology Associates Inc. (Columbus, OH). It was found to contain 97.3% molnupiravir, with no detection of other contaminants. The Aura 2801 product used by most of the participants in this study was analyzed in September 2022 by the same laboratory. It was found to contain 96.8% pure molnupiravir. A more controlled assessment of the true content and purity of the unlicensed formulations of both GS-441524 and molnupiravir is of great interest to the veterinary community and an active area of investigation for our group.

Some limitations of this study are due to the retrospective nature and the legality of the therapies being used. First, all data used for this study were owner-reported. Working closely with the owners and the social media website administrators who supported this cohort allowed for a greater understanding and interpretation of many survey responses. Due to the lack of definitive antemortem FIP diagnostics available for practical use, there was also an inability to confirm that the cats enrolled in this study had FIP. Moreover, the data are likely biased toward positive outcomes and may suffer from recall error. During the distribution phase, a possible study participant responded by asking to be removed from our email list and indicated they would not like to participate in the study. Their cat did not respond to molnupiravir treatment and was ultimately euthanized. It is hypothesized that others may have felt the same way, as three additional possible participants did not respond to the study invitation. This may limit the number of participants with unfavorable outcomes and falsely elevate the apparent survival. Thus, the data reported here are meant to serve as proof of the possibility of molnupiravir as a first-line or rescue therapy for FIP, not as a true rate of efficacy.

For the cats who used unlicensed molnupiravir as a rescue therapy, the cause of failure to respond or relapse following GS-441524-based therapy was not identified. It may have been related to drug quality, viral resistance, or some other factor. Since there is currently no testing or regulation in the USA, unlicensed versions of GS-441524 or GC376 may have insufficient purity or concentration, leading to treatment failure. Another possible cause is natural or acquired resistance to GS-441524. These two causes can also be related, as acquired resistance may be encouraged when an insufficient amount of the antiviral is used in therapy, such as with low-quality preparations of the drug.

A recent article found no SARS-CoV-2 drug-induced viral mutations during molnupiravir treatment [14]. This suggests that SARS-CoV-2 viral resistance is unlikely to emerge with molnupiravir therapy. Thus, molnupiravir therapy may be similarly unlikely to induce viral resistance in FIPV, making it an attractive therapeutic option.

However, there is clearly a need for (1) a legal (in the United States and elsewhere) alternative to unlicensed GS-441524 therapy and (2) the availability of alternative rescue drugs, either alone or in combination, following GS-441524 therapy failure. Molnupiravir has the potential to fill both of these niches, and this is the first known report of its use in cats in the scientific literature. Nevertheless, unlicensed products may continue to be used for FIP due to the cost and the widely established networks available to obtain them.

In conclusion, unlicensed molnupiravir appears to be an effective treatment for suspected FIP both as a first-line therapy and as a rescue therapy, according to owner-reported data. There are minimal reported side effects at the dosage of 12–15 mg/kg every twelve hours, and it provides the possibility of survival with the clinical remission of signs of FIP. Although unconventional and potentially unlawful, the experiences of these owners in treating and apparently curing cats of FIP is undeniably remarkable, and there is much we can learn from the experiments being conducted by these “citizen scientists.” By reporting these experiences, we aim to provide a starting point for the investigation of molnupiravir for use in cats with suspected FIP and to document a crowd-sourced health phenomenon that should not be ignored by our profession.

## Figures and Tables

**Figure 1 pathogens-11-01209-f001:**
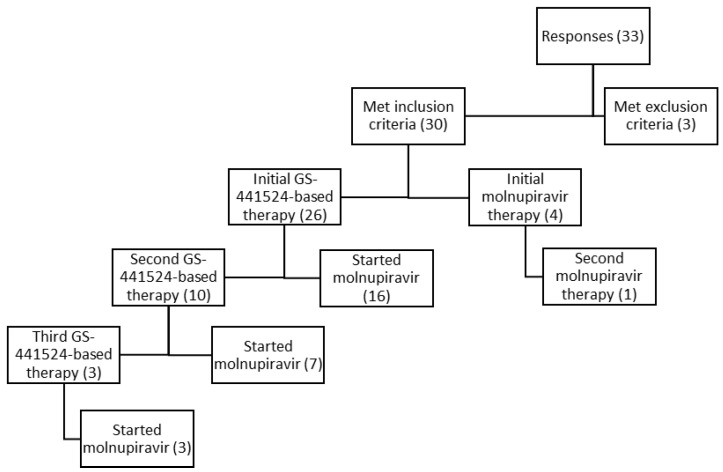
This flowchart depicts the number of cases in each treatment block.

**Figure 2 pathogens-11-01209-f002:**
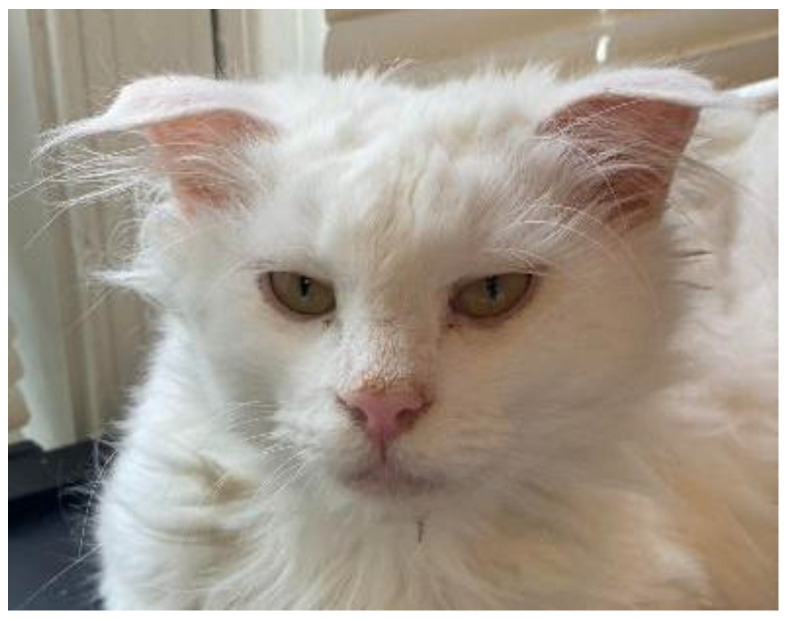
Folded ear tips were reported as an adverse effect of unlicensed molnupiravir therapy in Cat #21.

**Figure 3 pathogens-11-01209-f003:**
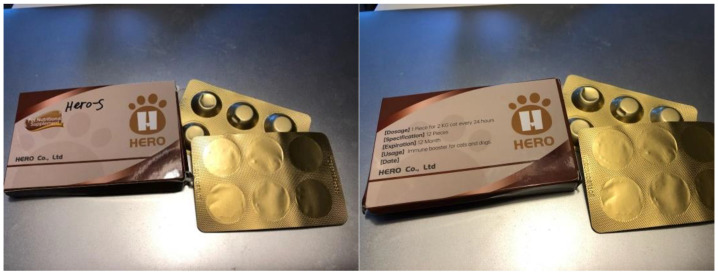
Images of packaging of Hero brand unlicensed molnupiravir product.

**Table 1 pathogens-11-01209-t001:** Signalment and characteristics of initial therapies for all 30 cats treated with unlicensed molnupiravir for suspected FIP.

Cat	Age at Diagnosis (Months)	Sex/Neuter Status at Diagnosis	Breed	Pre-Existing Medical Condititons	Country of Origin	Form of FIP	Initial Therapy	Initial Therapy Duration (Weeks)	Disease Free Interval	Second Therapy	Second Therapy Duration (Weeks)	Disease Free Interval	Third Therapy	Third Therapy Duration (Weeks)	Disease Free Interval
1	4	Male	European Shorthair	Parasitic infections, URI as kitten	Germany	Neurological	Injectable Oral GS-441524	8	None	Injectable and Oral GS-441524	15	None			
2	15	Spayed Female	Burmese	None	Sweden	Effusive, non-effusive, neurological	Injectable GS-441524	12	Less than 4 weeks	Injectable GS-441524	14	17 days	Oral GS-441524	5 weeks	None
3	9	Neutered Male	British Shorthair	None	Poland	Effusive, neurological, ocular	Injectable GS-441524	13	Less than 2 weeks	Injectable GS-441524	12	Longer than 6 months, less than 1 year			
4	5	Neutered Male	Abyssinian	None	USA	Effusive	Injectable GS-441524	12	Less than 2 weeks	Injectable GS-441524	14	Less than 4 weeks			
5	4	Spayed Female	Balinese Ragdoll mix	Calicivirus, conjunctivitis, giardiasis, ringworm, URI	USA	Non-effusive	Injectable GS-441524	13	Less than 8 weeks						
6	7	Spayed Female	Siamese	None	USA	Neurological	Injectable and oral GS-441524, injectable GC, injectable and oral molnupiravir	12	None						
7	7	Neutered Male	American Shorthair	None	USA	Non-effusive	Injectable and Oral GS-441524	5	None						
8	6	Spayed Female	American Shorthair/Siamese mix	Ringworm, FCoV	USA	Effusive, neurological	Injectable and Oral GS-441524	5	None						
9	4	Spayed Female	Domestic mixed breed	History of fractured pelvis	USA	Effusive	Injectable and Oral GS-441524	14	Less than 6 months	Oral GS-441524	13	Less than 4 weeks	Oral GS-441524/injectable GC	6 weeks in combination then 6 weeks of oral GS	None
10	4	Neutered Male	Domestic mixed breed	None	USA	Effusive	Injectable GS-441524	23	Less than 4 weeks						
11	72	Neutered Male	Domestic mixed breed	FeLV	USA	Non-effusive	Oral GS-441524	12	Less than 6 months						
12	5	Male	Domestic mixed breed	None	USA	Non-effusive, neurological, ocular	Injectable and oral GS-441524	17	None						
13	1.5	Male	Savannah	None	USA	Effusive, neurological	Injectable and oral GS-441524	24	Less than 6 months	Injectable and Oral GS-441524	12	Less than 4 weeks			
14	4	Spayed Female	Domestic mixed breed	Skin and eye infections, fleas	poland	Non-effusive, neurological	Injectable GS-441524	12	Less than 2 weeks	Injectable GS-441524	17	Less than 4 weeks			
15	12	Spayed Female	American Shorthair	None	USA	Effusive	Injectable GS-441524/GC	1.5	None						
16	5	Spayed Female	Domestic mixed breed	None	USA	Effusive, neurological	Injectable GS-441524	12	Less than 4 weeks						
17	4	Male	American Longhair	None	USA	Ocular	Injectable and Oral GS-441524, GC376	13	None						
18	6	Neutered Male	Domestic mixed breed	None	USA	Effusive	Injectable GS-441524	12	None						
19	12	Neutered Male	Domestic mixed breed	None	USA	Non-effusive	Injectable and Oral GS-441524	12	Less than 2 weeks	Injectable GS-441524	12	None			
20	6	Neutered Male	Unknown	None	USA	Non-effusive, neurological	Injectable GS-441524	4	None	Oral GS-441524	3	None			
21	4	Spayed Female	Norwegian Forest Cat	None	USA	Neurological	Injectable GS-441524	12	Less than 6 months	Injectable GS-441524	1.5	None	Molnupiravir, GS-441524, GC	12 weeks	None
22	6	Neutered Male	Domestic mixed breed	None	USA	Neurological, ocular	Oral GS-441524	3	None						
23	12	Spayed Female	Unknown breed	None	Germany	Neurological	Injectable GS-441524	16	Less than 6 months						
24	3	Male	Domestic mixed breed	None	USA	Neurological	Injectable GS-441524	12	Less than 6 months						
25	6	Neutered Male	American Shorthair	None	USA	Effusive	Oral GS-441524	13	Less than 1 week						
26	1	Male	Unknown breed	None	USA	Non-effusive	Injectable GS-441524	12	Less than 1 week						
27	7	Neutered Male	Domestic mixed breed	None	USA	Non-effusive, neurological	Molnupiravir	12	Less than 1 week	*Molnupiravir					
28	24	Spayed Female	Domestic mixed breed	None	USA	Effusive	Molnupiravir								
29	12	Spayed Female	Domestic mixed breed	None	USA	Non-effusive, ocular	Molnupiravir								
30	24	Neutered Male	Domestic mixed breed	None	USA	Neurological	Molnupiravir								

**Table 2 pathogens-11-01209-t002:** Treatment and outcome characteristics of 26 cats that received unlicensed molnupiravir as a rescue therapy.

Cat	Clinical Signs at Start of Treatment	Brand Name	Starting Dose and Frequency	Ending Dose and Frequency	Duration of Treatment (Weeks)	Time to Improvement	Persistent Clinical Signs	Outcome	Adverse Effects
1	Diarrhea, vomiting	Aura Plus	11 mg/kg twice a day	11 mg/kg twice a day	12	Less than 1 week	None	Clinical Remission	None
2	None reported	Aura	12 mg/kg twice a day	12 mg/kg twice a day	12	Unsure	None	Clinical Remission	None
3	Anisocoria, color spots in the eye, polydypsia, pica, weight loss	Aura 2801	28 mg/kg twice a day	14 mg/kg twice a day	12	Within 2 weeks	None	Clinical Remission	None
4	Anorexia, lethargy, weight loss	EIDD	7 mg/kg twice a day	7 mg/kg twice a day	12	Less than 1 week	None	Clinical Remission	None
5	Color spots in the eye, diarrhea, hiding and lack of socialization	Aura 2801	6 mg/kg once a day	13 mg/kg once a day	10	Within 2 weeks	None	Clinical Remission	None
6	Anisocoria, constipation, anorexia, fecal and urinary incontinence, lethargy, paralysis, seizures, pale gums, weight loss	Aura 2801	20 mg/kg twice a day	20 mg/kg twice a day	11	Less than 1 week	None	Clinical Remission	None
7	Anorexia, difficulty walking, hiding, lack of socialization, jaundice, lethargy	Capella EIDD	9 mg/kg twice a day	13 mg/kg twice a day	10	Less than 1 week	None	Clinical Remission	None
8	Anorexia, difficulty walking, urinary incontinence, paralysis	Aura 2801	17 mg/kg twice a day	17 mg/kg twice a day	15	Less than 1 week	Difficulty walking persisted for 2 months, still not normal but has normal life	Clinical Remission	None
9	Cough, anorexia, hiding, lack of socialization, polydypsia, weight loss	Aura 2801	12 mg/kg twice a day	16 mg/kg twice a day	13	Within 2 weeks	Polydypsia persisted for 1 week	Clinical Remission	None
10	Anorexia, lethargy, weight loss	Aura 2801	12 mg/kg twice a day	12 mg/kg twice a day	16	Within 2 weeks	None	Clinical Remission	None
11	Anorexia, lethargy, URI, weight loss	Aura 1931	12 mg/kg twice a day	12 mg/kg twice a day	12	Within 2 weeks	None	Clinical Remission	None
12	Blindness, head bobbing, difficulty walking	Aura 2801	10 mg/kg twice a day	14 mg/kg twice a day	12	Within 3 weeks	None	Clinical Remission	None
13	Difficulty walking, hiding, lack of socialization, polyuria, lethargy, anorexia, paralysis, tremors	Aura 2801	12 mg/kg twice a day	12 mg/kg twice a day	12	Less than 1 week	None	Clinical Remission	None
14	Anorexia, difficulty walking, hiding, lack of socialization, lethargy, unusually fearful	Aura 2801	11 mg/kg twice a day	16 mg/kg twice a day	18	Greater than 4 weeks	Nothing physical but the MRI is still not normal	Clinical Remission	None
15	Blindness, constipation, anorexia, diarrhea, distended abdomen, hiding, lack of socialization, lethargy, pale gums, weight loss	Aura 2801	16 mg/kg twice a day	16 mg/kg twice a day	12	Less than 1 week	None	Clinical Remission	None
16	Anorexia, difficulty walking, lethargy, seizures, tremors, weight loss	Aura 2801	14 mg/kg twice a day	14 mg/kg twice a day	12	Less than 1 week	None	Clinical Remission	None
17	Cough, anorexia, difficulty breathing, hiding, lack of socialization, lethargy, vomiting, weight loss	Aura 2801 and Aura 1931	12 mg/kg twice a day	17 mg/kg twice a day	20	Within 3 weeks	Anorexia	Clinical Remission	Nausea/vomiting, anorexia
18	Constipation, anorexia, difficulty walking, hiding, lack of socialization, weight loss	Aura 2801	12 mg/kg twice a day	12 mg/kg twice a day	8	Within 2 weeks	None	Clinical Remission	None
19	Lethargy, anorexia	Aura 2801	12 mg/kg twice a day	12 mg/kg twice a day	7	Within 2 weeks	None	Clinical Remission	None
20	Tremors/shaking	Aura 2801	10 mg/kg twice a day	23 mg/kg two-three times per day	10	Less than 1 week	In remission for approximately 1 week before seizures started	Euthanized	Decreased appetite at three times per day dosing, severe leukopenia, losing whiskers, flaky skin on ears
21	Difficulty walking, fecal incontinence	Aura 2801 and Aura 1931	13 mg/kg twice a day	30 mg/kg twice a day	14	Less than 1 week	Difficulty walking, difficulty jumping, fecal incontinence persisted at time of study (1 week post treatment)	Relapsed and Euthanized	Folded ear tips, muscle wasting
22	Color spots in the eye, anorexia, difficulty walking, hiding, lack of socialization, lethargy	Aura 2801	16 mg/kg twice a day	19 mg/kg twice a day	9	Within 2 weeks	None	Clinical Remission	None
23	Difficulty walking, anorexia, loss of balance	Aura EIDD	12 mg/kg twice a day	15 mg/kg three times a day	10	Within 2 weeks	Difficulty walking	Clinical Remission	None
24	Blindness, color spots in the eyes, anorexia, difficulty breathing, difficulty walking, distended abdomen, urinary incontinence, jaundice, lethargy, paralysis, tremors	Aura 2801	15 mg/kg twice a day	15 mg/kg twice a day	16	Less than 1 week	None	Clinical Remission	None
25	Difficulty breathing, difficulty walking, hiding, lack of socialization, lethargy, URI	Aura2801	7 mg/kg twice a day	7 mg/kg twice a day	16	Within 2 weeks	None	Clinical Remission	None
26	Lethargy, anorexia	Aura 2801	14 mg/kg twice a day	14 mg/kg twice a day	15	Less than 1 week	Neuological twitching, increased liver enzymes	Clinical Remission	None

**Table 3 pathogens-11-01209-t003:** Treatment and outcome characteristics of 4 cats who received unlicensed molnupiravir as first-line therapy.* Received two rounds of molnupiravir therapy; first round documented in Table 1.

Cat	Clinical Signs at Start of Treatment	Brand Name	Starting Dose and Frequency	Ending Dose and Frequency	Duration of Treatment (Weeks)	Time to Improvement	Persistent Clinical Signs	Outcome	Adverse Effects
* 27	Hiding, lack of socialization, lethargy, anorexia, URI, vomiting, weight loss	Aura 2801	19 mg/kg twice a day	19 mg/kg twice a day	10	Less than 1 week	None	Clinical Resmission	None
28	Anoreixa, difficulty walking, distended abdomen, hiding, lack of socialization, lethargy	Aura 2801	8 mg/kg twice a day	8 mg/kg twice a day	13	Within two weeks	None	Clinical Resmission	None
29	Anisocoria, blindness, color changes in eyes, anorexia, hiding, lack of socialization, urinary incontinence, lethargy,	Aura 2801	10 mg/kg twice a day	10 mg/kg twice a day	13	Within 2 weeks	None	Clinical Resmission	None
30	Hiding, lack of socialization, lethargy, pale gums, weight loss	Aura 2801	10 mg/kg twice a day	12 mg/kg twice a day	10	Within two weeks	None	Clinical Resmission	None

## Data Availability

The data not included within the tables presented in this study are available on request from the corresponding author.

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
