# Peer review of "Unlicensed Molnupiravir is an Effective Rescue Treatment Following Failure of Unlicensed GS-441524-like Therapy for Cats with Suspected Feline Infectious Peritonitis"

_pathogens, 2022, doi:10.3390/pathogens11101209_

Round 1

Reviewer 1 Report

This is the second paper from Samantha Evans group. It uses a survey to document treatment of cats with FIP with molnupiravir. The subject is important. The research design has lots of inherent problems – as the authors admit to selection bias (beyond their control), the data set is incomplete, the cases are very different and treatment regimens are complex. It would have been much better if all the cases had been treated at OSU! Never-the-less – there is compelling data that is some/most cats, molnupiravir at a dose rate of roughly 10-15 mg/kg is effective at treating  (1) new cases of FIP (numbers small) (2) and recurrent cases that failed treatment with GS-441524/GC376. So – these are the hardest cases to treat, the worst of the worse, where presumably the FIP virus has developed mutational resistance to GS-44152. Treatment  seems successful, and toxicity is mild or absent – even though who know how many cases had careful haematological and biochemical monitoring. So, on this basis alone – the work deserves publication.

Issues

1.       The authors talk about duration of treatment for the study cohort in terms or average. I think much better to describe duration of treatment with median and giving the IQR. I don’t think it will change much.

2.       The authors sometimes use the words symptoms when really the correct term would be signs or clinical signs

3.       Most cases were treated with molnupiravir, but some were treated with AURA 1931 which I think refers to the active drug molnupiravir gets turned into viz EIDD-1931. This means these cats were treated at a different nominal dose rates as the molecular weight of  1931 is lower than molnupiravir, so at the nominal dose rate – they were giving MORE of the active drug. This needs clarification. Were these cats different – did they get better faster?

4.       The authors have little control over the data – they are reliant on the memory of the  participants and the records they kept. But the authors CAN check the reliability of the medications that were given, and this needs clarification before the paper is accepted for publication. The authors state they have tested one sample of molnupiravir for purity – it was 97%. What we need to know – is how  much of the active drug was present in each tablet. The authors need to buy some HERO 2801, AURA 2801 and Aura 1931 – and have them assayed for  molnupiravir and EIID 1931 – and enough tablets from different batches that they are confident of how much of which drug is present in the tablet. I am only suggesting perhaps 10-20 tablets need to be assayed – just to get an impression of the coefficient of variation of active drug in the tablets people were using. This should be done very quickly – as it would be great to get this paper in the literature.

5.       Finally – in the Discussion the authors raise the tricky issue about whether this study is legal. They had the same issue in the Animals paper on GS-441524. I don’t actually mind whether its legal or not, but I know many clinicians in the USA do worry. The difference between molnupiravir and GS-441524 – is that the former has FDA approval. So, any veterinarian in the USA can write a prescription for molnupiravir – get a course of therapy (40 200 mg capsules – cost 1100 dollars in Australia where  live), get it compounded to the correct strength for a cat, and treat the cat off label with a licensed human drug (get the owner to sign a ware etc). Then we know for sure about the quality of active in the medication. Now I realise it’s all about cost, and the huge advantage of molnupiravir is that as well as being licensed, is its far cheaper than GS-441524 ($209 dollars for the unlicensed stuff versus $3448.83 for unlicensed GS-441524), but there is the option of importing human grade molnupiravir from India (where it’s made under license) for 50 dollars (!!), which then could be compounded to the correct strength. I know how to do this in Australia – but our laws are different to the USA and Europe. But I have advised many vets in India, South Africa about using molnupiravir – and the human stuff is widely available and cheap in many many countries. I think the authors might investigate the legality of this, and also work out the cost of using off label human molnupiravir in the USA – as I know this is legal and i know it will only get cheaper as COVID  gets more and more under control. I think this is exciting and important, as this can be legal – which means vets can start to be involved more in therapy, and its far cheaper than GS-441524. Will it be just as good? – time will tell.

Author Response

Dear reviewer - thank you for your time and expertise in helping us improve our manuscript.  We have added a point-by-point response below.  Each of our responses are found in BLUE font and start with an asterisk (*).

REVIEWER 1

This is the second paper from Samantha Evans group. It uses a survey to document treatment of cats with FIP with molnupiravir. The subject is important. The research design has lots of inherent problems – as the authors admit to selection bias (beyond their control), the data set is incomplete, the cases are very different and treatment regimens are complex. It would have been much better if all the cases had been treated at OSU! Never-the-less – there is compelling data that is some/most cats, molnupiravir at a dose rate of roughly 10-15 mg/kg is effective at treating  (1) new cases of FIP (numbers small) (2) and recurrent cases that failed treatment with GS-441524/GC376. So – these are the hardest cases to treat, the worst of the worse, where presumably the FIP virus has developed mutational resistance to GS-44152. Treatment  seems successful, and toxicity is mild or absent – even though who know how many cases had careful haematological and biochemical monitoring. So, on this basis alone – the work deserves publication.

Issues

  1.       The authors talk about duration of treatment for the study cohort in terms or average. I think much better to describe duration of treatment with median and giving the IQR. I don’t think it will change much.

*Thank you for this suggestion - the IQR has been calculated and added into the manuscript for all treatment durations. The only exception being for the 2 ocular cases as the IQR for 2 data points cannot be calculated.

  1.       The authors sometimes use the words symptoms when really the correct term would be signs or clinical signs. 

*Thank you for recognizing this error - it has been changed throughout the manuscript and in Tables 2 & 3. 

  1.       Most cases were treated with molnupiravir, but some were treated with AURA 1931 which I think refers to the active drug molnupiravir gets turned into viz EIDD-1931. This means these cats were treated at a different nominal dose rates as the molecular weight of  1931 is lower than molnupiravir, so at the nominal dose rate – they were giving MORE of the active drug. This needs clarification. Were these cats different – did they get better faster?

*Thank you for this important point - the manuscript has been edited to reflect this concern.  

  1.       The authors have little control over the data – they are reliant on the memory of the  participants and the records they kept. But the authors CAN check the reliability of the medications that were given, and this needs clarification before the paper is accepted for publication. The authors state they have tested one sample of molnupiravir for purity – it was 97%. What we need to know – is how  much of the active drug was present in each tablet. The authors need to buy some HERO 2801, AURA 2801 and Aura 1931 – and have them assayed for  molnupiravir and EIID 1931 – and enough tablets from different batches that they are confident of how much of which drug is present in the tablet. I am only suggesting perhaps 10-20 tablets need to be assayed – just to get an impression of the coefficient of variation of active drug in the tablets people were using. This should be done very quickly – as it would be great to get this paper in the literature. 

*Our original molnupiravir analysis was on HERO brand, however, we recently also received results for Aura 2801 analysis, which has been added to the manuscript. Given the rapid turn-around time for these revisions (5 days), additional analyses are not feasible.  However, as mentioned in the manuscript, we agree that this (as well as analysis of unlicensed GS-441524 compounds) is a very important avenue of further research, for both safety and efficacy reasons.  Thus, we plan to complete the described analysis as soon as we can obtain funding (we largely already have donated compounds) to do so.  

  1.       Finally – in the Discussion the authors raise the tricky issue about whether this study is legal. They had the same issue in the Animals paper on GS-441524. I don’t actually mind whether its legal or not, but I know many clinicians in the USA do worry. The difference between molnupiravir and GS-441524 – is that the former has FDA approval. So, any veterinarian in the USA can write a prescription for molnupiravir – get a course of therapy (40 200 mg capsules – cost 1100 dollars in Australia where  live), get it compounded to the correct strength for a cat, and treat the cat off label with a licensed human drug (get the owner to sign a ware etc). Then we know for sure about the quality of active in the medication. Now I realise it’s all about cost, and the huge advantage of molnupiravir is that as well as being licensed, is its far cheaper than GS-441524 ($209 dollars for the unlicensed stuff versus $3448.83 for unlicensed GS-441524), but there is the option of importing human grade molnupiravir from India (where it’s made under license) for 50 dollars (!!), which then could be compounded to the correct strength. I know how to do this in Australia – but our laws are different to the USA and Europe. But I have advised many vets in India, South Africa about using molnupiravir – and the human stuff is widely available and cheap in many many countries. I think the authors might investigate the legality of this, and also work out the cost of using off label human molnupiravir in the USA – as I know this is legal and i know it will only get cheaper as COVID  gets more and more under control. I think this is exciting and important, as this can be legal – which means vets can start to be involved more in therapy, and its far cheaper than GS-441524. Will it be just as good? – time will tell. 

*Thank you very much for this added insight.  Currently molnupiravir is only approved under an Emergency Use Authorization for human COVID-19 patients in the USA. Thus it cannot be prescribed off label by MDs or DVMs. The hope is that soon it will receive full FDA approval and DVMs will be able to start prescribing the medication as they do in other countries. But you are correct that we predict that the underground market will continue regardless, mostly related to cost. We attempted to elaborate on this topic in the discussion of the manuscript. 

Reviewer 2 Report

This is an interesting study. In the UK we also feel molnupiravir may be helpful for cases relapsing or neurological cases failing elevations in GS441524 dose. I have mixed feelings about potentially promoting illegal products but recognise in the USA no legal products are available as we have now in the UK. I think this article adds to the general literature on this topic. 

In figure 1 the first box is followed by a box of 30 'Met inclusion criteria' and 3 'met inclusion criteria' - do you mean did not meet inclusion criteria for the 3 cats?

line 149 typo in GS441524

Can you insert more information on the leukopenia - the discussion states you had this information from the vets - neutropenia? What level?

I can't see the tables - not included in submission?

Author Response

Dear reviewer - thank you for your time and expertise in helping us improve our manuscript.  We have added a point-by-point response below.  Each of our responses are found in BLUE font and start with an asterisk (*).

REVIEWER 2

This is an interesting study. In the UK we also feel molnupiravir may be helpful for cases relapsing or neurological cases failing elevations in GS441524 dose. I have mixed feelings about potentially promoting illegal products but recognise in the USA no legal products are available as we have now in the UK. I think this article adds to the general literature on this topic. 

In figure 1 the first box is followed by a box of 30 'Met inclusion criteria' and 3 'met inclusion criteria' - do you mean did not meet inclusion criteria for the 3 cats?

*Thank you for pointing this out. The box on the right actually says “3 Met EXclusion criteria.” If this is confusing wording, we can change it to “3 did not meet inclusion criteria.”

line 149 typo in GS441524 

*Thank you for pointing this out; it has been corrected.

Can you insert more information on the leukopenia - the discussion states you had this information from the vets - neutropenia? What level?

*Thank you for this important point - the manuscript has been edited in the results section to reflect this concern.

I can't see the tables - not included in submission?

*The tables were included as “supplemental” data due formatting issues (these are very large tables that are best viewed in a horizontal or landscape orientation).  We will leave it up to the copy editors as to whether they can be formatted to be included in the published manuscript, or as supplemental data.  But the reviewer should have access to this currently.

Reviewer 3 Report

Thank you for a valuable submission, and this on the balance is a well written, well planned study. You have done a responsible job of retrieving credible owner reported information, and of revealing the caveats. My guess would be that this group of devoted owners would not be avoiding reporting failures, and the very study design here would weed out any answers mis-reported due to laziness. I really like the "breakdown" in analysis of effectiveness (different forms of FIP) although I would be surprised if any of the numbers can be ignificant without further cases. I also appreciate the effort to report on co-morbidities or co-infections.

Thank you for providing the 58 page questionnaire. I would suggest that the authors provide more, and more specific background about the scope and nature of the study.  There is a good deal of information, in addition to questions about Molnupiravir, asked on the survey that is related to diagnosis of FIP, clinical signs, presentation of FIP, Remdesevir, and GS-441524. While some of this was inclusion and exclusion criteria, a good deal was not reported in this paper. Tables not provided (to this reviewer) either with the supplemental materials nor embedded in in the paper, so that will be a problem for the completeness of this review, and forgive me if this information IS provided in Table 1.

Could the authors elaborate on why the focus on the effectiveness of molnupiravir rather than effectiveness of combos or a focus on eg GS-441524 or remdesevir? Does it all come down to liscencing? Is access to molnupiravir license relatively facile?

The math leading up to the 30 participating cats was confusing (lines 93-103), so the flow chart greatly appreciated. It is impressive that among 37 requests to fill out a 58 page survey including requests for materials, 30 responses were logged. Some of the information reported here is vague (17 owners attached "documents"). If that information changed the inclusion exclusion criteria.. it is relevant, but otherwise does not add to the flow of the case definition.

Is it reasonable to ask or comment on the sources for these drugs? This is partially provided for monlupiravir, but not for other drugs: specifically to comment on how likely it would be that lack of response might be related to drug quality? Forgive my ignorance on the availability and quality of the therapy.

lines 295- When you discuss the difference in treatments performed by these owners (against the HERO company recommendations) it would be useful to suggest why treatments were done against suggested protocol? or whether your study verified that different forms of FIP would potentially be curable with different doses.

Author Response

Dear reviewer - thank you for your time and expertise in helping us improve our manuscript.  We have added a point-by-point response below.  Each of our responses are found in BLUE font and start with an asterisk (*).

REVIEWER 3

Thank you for a valuable submission, and this on the balance is a well written, well planned study. You have done a responsible job of retrieving credible owner reported information, and of revealing the caveats. My guess would be that this group of devoted owners would not be avoiding reporting failures, and the very study design here would weed out any answers mis-reported due to laziness. I really like the "breakdown" in analysis of effectiveness (different forms of FIP) although I would be surprised if any of the numbers can be significant without further cases. I also appreciate the effort to report on co-morbidities or co-infections.

Thank you for providing the 58 page questionnaire. I would suggest that the authors provide more, and more specific background about the scope and nature of the study.  There is a good deal of information, in addition to questions about Molnupiravir, asked on the survey that is related to diagnosis of FIP, clinical signs, presentation of FIP, Remdesevir, and GS-441524. While some of this was inclusion and exclusion criteria, a good deal was not reported in this paper. Tables not provided (to this reviewer) either with the supplemental materials nor embedded in in the paper, so that will be a problem for the completeness of this review, and forgive me if this information IS provided in Table 1.

*This retrospective cohort study sampled 37 subjects from a sample pool of 80 potential participants, who belong to an online support community for treating FIP. Participants were recruited into the study through email invitation and were followed-up a total of three times through reminder emails. We formatted the survey instrument using demographic and clinical survey questions from previous studies to remain consistent in language and style. The survey instrument was constructed to be completed in a one-time sitting and used conditional-logic question formatting frequently to reduce reporting bias and survey fatigue by the participants. Additionally, because this study is retrospective, we attempted to limit the number of free-response questions asked in the survey, so as to not induce recall bias. 

*The tables were included as “supplemental” data due formatting issues (these are very large tables that are best viewed in a horizontal or landscape orientation).  We will leave it up to the copy editors as to whether they can be formatted to be included in the published manuscript, or as supplemental data.  But the reviewer should have access to this currently. The tables do include more information than what is reported in the manuscript, including signalment and clinical signs before starting molnupiravir. The authors believe all important information was reported in the study.

Could the authors elaborate on why the focus on the effectiveness of molnupiravir rather than effectiveness of combos or a focus on eg GS-441524 or remdesevir? Does it all come down to liscencing? Is access to molnupiravir license relatively facile? 

*We chose to focus on the effectiveness of molnupiravir because it had not yet been reported in the literature and we decided that documenting the use of molnupiravir as an FIP therapy was very important.  Molnupiravir is also on the horizon for being FDA approved in the USA and a documented therapeutic protocol is needed. Additionally, there is published  literature discussing the efficacy of GS-441524 and GC-376, and a prospective study currently being conducted on at-home GS-441524 therapy by several of the authors. 

The math leading up to the 30 participating cats was confusing (lines 93-103), so the flow chart greatly appreciated. It is impressive that among 37 requests to fill out a 58 page survey including requests for materials, 30 responses were logged. Some of the information reported here is vague (17 owners attached "documents"). If that information changed the inclusion exclusion criteria.. it is relevant, but otherwise does not add to the flow of the case definition.

*Thank you for pointing this out. The medical history and other documents uploaded were only used as supporting evidence for the adverse reactions reported by the participants. This information was also collected and stored for potential future studies. 

Is it reasonable to ask or comment on the sources for these drugs? This is partially provided for monlupiravir, but not for other drugs: specifically to comment on how likely it would be that lack of response might be related to drug quality? Forgive my ignorance on the availability and quality of the therapy.

*Thank you for this suggestion- An explanation has been provided in the discussion of the manuscript. 

lines 295- When you discuss the difference in treatments performed by these owners (against the HERO company recommendations) it would be useful to suggest why treatments were done against suggested protocol? or whether your study verified that different forms of FIP would potentially be curable with different doses.

*Many of the cats in this study used molnupiravir from another supplier, Aura, who did not make specific recommendations as to the treatment protocol, and the cat owners instead relied on the white paper published by Dr. Pedersen and Nicole Jacque, as well as advice from cat owner groups on social media.  While the Hero dosing information was known within these groups, there was skepticism about their results, as well as anecdotal stories of cats having adverse reactions.  These cats were following advice from advisors in the social media groups to use the lowest dosage that seemed effective, starting with the dosages published in the whitepaper and titrating upwards as necessary.  Some dosing was also decided by the practicality of how 50 mg tablets could reasonably be divided. 

Round 2

Reviewer 1 Report

Excellent revision. Good to go!

Reviewer 3 Report

The responses to the reviewer were thoughtful. The paper is very useful for clinicians.

Some of the information supplied only as a reply to the reviewer could have been added to the paper (esp the Aura supplier lack of protocol and white paper use etc).